# Grating-incoupled waveguide-enhanced Raman sensor

**Mohamed A. Ettabib** [1]*, **Bethany M. Bowden**[2], **Zhen Liu**[1], **Almudena Marti**[2], **Glenn M. Churchill** [1], **James C. Gates**[1], **Michalis N. Zervas**[1], **Philip N. Bartlett**[2], **James S. Wilkinson**[1]

**1** Zepler Institute for Photonics and Nanoelectronics, University of Southampton, Southampton, United Kingdom, **2** School of Chemistry, University of Southampton, Southampton, United Kingdom

* m.ettabib@soton.ac.uk

**Data Availability Statement:** All data supporting this study are openly available at https://doi.org/10.5258/SOTON/D1855.

**Funding:** This work was funded by a grant from the UK Engineering and Physical Sciences Research

## Abstract

We report a waveguide-enhanced Raman spectroscopy (WERS) platform with alignment-tolerant under-chip grating input coupling. The demonstration is based on a 100-nm thick planar (slab) tantalum pentoxide ($Ta_2O_5$) waveguide and the use of benzyl alcohol (BnOH) and its deuterated form (d7- BnOH) as reference analytes. The use of grating couplers simplifies the WERS system by providing improved translational alignment tolerance, important for disposable chips, as well as contributing to improved Raman conversion efficiency. The use of non-volatile, non-toxic BnOH and d7-BnOH as chemical analytes results in easily observable shifts in the Raman vibration lines between the two forms, making them good candidates for calibrating Raman systems. The design and fabrication of the waveguide and grating couplers are described, and a discussion of further potential improvements in performance is presented.

## 1. Introduction

Waveguide enhanced Raman spectroscopy (WERS) is a promising tool for enabling the detection and analysis of biochemical species by measuring their unique Raman spectra. WERS relies on the evanescent excitation of the target molecules and the evanescent collection of the Raman signal through waveguide modes [1]. Recent advances in nanofabrication techniques, material engineering, optoelectronic instrumentation and software modelling tools have resulted in substantial improvements in the Raman conversion efficiency of WERS systems compared to earlier demonstrations [2, 3], and to significantly stronger Raman signals than those collected by conventional Raman microscopes [4]. Further, WERS operates on the same principles as surface-enhanced Raman spectroscopy (SERS) with electromagnetic enhancement of Raman scattering at the sensing surface, and is thus expected to be useful in the same range of analytical applications as SERS [1], where it would be would be applicable to a wide range of biosensor assays, such as lateral flow immunoassays, DNA discrimination and detection, affinity assays, and biomarker detection, in the same way as SERS [5, 6]. Similar approaches would also be adopted for the suppression of interferences such as surface modification to suppress non-specific adsorption [7]. However, WERS sensors offer a number of advantages over the more mature SERS technology, as they are low cost, offer a high degree of

Council (EPSRC), Grant Number EP/R011230/1. B.
M.B. thanks the Defence Science and Technology
Laboratory (contract no. DSTLX-1000128554) for
supporting an EPSRC industrial CASE award. the
funders had no role in study design, data collection
and analysis, decision to publish, or preparation of
the manuscript.

**Competing interests:** The authors have declared
that no competing interests exist.

miniaturization, and therefore enhanced practicality and usability, and unlike SERS, do not
utilize expensive and fragile noble metal nanostructures. Furthermore, unlike SERS, the optical
excitation and collection light-paths do not pass through the liquid sample volume, reducing
interference from absorption and scattering, also removing the need for an optical window for
light to enter and leave the sample and allowing direct placement of fluidics on the waveguide.
Further, both polarisations are available for excitation and collection, providing complementary Raman information. These advantages have recently led to a series of useful demonstrations, ranging from the detection of monolayers [8], trace gases [9, 10] and explosives [11] to
the first packaged fiber-coupled WERS sensor [12].

Despite this rapid progress, there remain some challenges facing the wider adoption of
WERS technology, such as (i) high waveguide propagation losses, which limit the maximum
effective waveguide length over which to build the Raman signal [13] and (ii) high laser input
coupling losses, which limit system efficiency. The high coupling losses are partly due to the
nature of the ultrathin film planar waveguides (<150 nm) and nanoscale channel waveguides
that are usually used in WERS devices, which make efficient in- and out-coupling of light a
challenging process for disposable devices in the field, where end-coupling to ultrathin high
index contrast waveguides is made difficult by their small mode size (~1 μm FWHM) which
requires a similar incident spot/mode for good coupling efficiency and submicron alignment
tolerances, or alternatively integrated mode transformers which still demonstrate micron-scale
alignment tolerances. Recent WERS demonstrations have so far relied either on end-fire coupling using aspheric lenses, microscope objective lenses or monomode fibers or prism coupling, with minimum coupling losses ranging from 3 dB [14] to 8 dB [8]. The use of end-fire
(or edge) coupling requires very precise alignment and end-facet cleaving and polishing or
deep plasma etching while prism coupling requires fine adjustment of the coupling strength
and is incompatible with monolithic integration.

An alternative approach to light coupling into WERS chips is the use of monolithically integrated grating couplers. Grating couplers provide better translational alignment tolerances
than end-facet coupling [15, 16], particularly for the ultrathin waveguides. Furthermore, they
can be freely located at any position on the chip surface, allow launching from either above or
underneath the chip, and relax the requirement for precise facet polishing. In addition, their
most significant drawback, namely their narrow bandwidth, is not problematic for pump
input coupling in WERS applications since mode excitation is achieved with a narrow linewidth laser. Such gratings can also be readily designed to pass the Stokes-shifted Raman signal,
thus facilitating backward collection from the end facet if required. Gratings are not required
for out-coupling of light from the chip, as out-coupling is a straightforward alignment-tolerant
process that can be implemented using a large core multimode fiber butt-coupled to the end
facet of the waveguide to collect the Raman emission, with no need for any alignment control
(due to the large dimensions of the collection fiber core). For example, the translational alignment tolerance (defined as 3dB reduction in coupling efficiency) for coupling from an ultra-
thin slab waveguide into a 1mm core diameter high NA fibre is estimated to be ±0.4 mm.

While grating couplers are extensively used in silicon photonics for coupling light into
channel waveguides, alignment tolerances are greatly relaxed for the excitation of slab waveguides, and while these demonstrate slightly lower WERS enhancement per unit length compared with channel waveguides [14], they are simpler to fabricate and have no additional
waveguide loss contribution from sidewalls. Whereas the use of spirals, possible with channel
waveguides, can lead to longer waveguides and thus an enhanced WERS signal, such an
approach has proven difficult to exploit due to the high waveguide losses which limit the maximum useful waveguide length. Slab waveguides have the added potential advantage of being
compatible for use in polarization-resolved studies of the orientation of highly ordered

**Table 1. A comparison of the main coupling techniques, in terms of efficiency, bandwidth, alignment tolerance, cost and complexity, that are used for WERS devices.** Quoted coupling efficiency values correspond to those reported in the literature for WERS.

| | Coupling efficiency | Bandwidth | Alignment Tolerance | Cost | Complexity |
|---|---|---|---|---|---|
| Prism coupling | 10–30% | Moderate; tens of nanometres | Good; tens of microns | Moderate | Moderate. Requires prism attachment |
| Grating couplers | 30–50% | Moderate; tens of nanometres | Good; tens of microns | Moderate | Moderate. Requires a nanofabrication facility |
| Aspheric lenses / Microscope objective lenses | 10–30% | High; hundreds of nanometres. | Poor; a few microns | Moderate | Easy to set-up |
| Fiber pigtailing with inverse tapers | 50% | High; hundreds of nanometres | Good—once fibres are fixed to the chip | High | Complex. Requires a nanofabrication facility and precise active alignment and attachment |

molecular layers. Table 1 below summarizes the main differences between the various coupling methodologies used for WERS sensors.

We have recently presented simulations that demonstrate that high coupling efficiencies can be obtained using simple grating couplers in most of the commonly-used WERS materials platforms [17]. Moreover, we introduced a new design figure-of-merit (FOM) that combines the grating's coupling efficiency optimization with the surface intensity optimization, since both processes are a function of film thickness. In this paper, we build on this theoretical work by experimentally demonstrating a $Ta_2O_5$ waveguide chip with a grating coupler designed for excitation through the substrate from the underside of the chip. This approach offers good translational alignment tolerance and frees up the top of the sample for sensing. The WERS system is demonstrated by measuring the Raman spectrum of benzyl alcohol (BnOH) and its deuterated form d7-BnOH, which show promise as standard analytes for calibrating Raman systems, since they possess comparable Raman cross-sections to toluene [18], the current commonly used standard analyte, while being non-volatile and having low toxicity.

## 2. Waveguide design and fabrication

For a low-index contrast waveguide, it is sufficiently accurate to select the optimum film thickness to maximize the surface intensity when designing a slab waveguide for a WERS process [19]. However, for a high-index contrast waveguide, the presence of a significant non-zero longitudinal electric field component for the TM mode means the optimization should be carried out in terms of the square of the magnitude of the surface total electric field ($|E|^2$), rather than the surface intensity [2] as the evanescent electric field may contain a component in the direction of modal propagation. In addition to this, it is important to combine the optimization of the excitation and collection of the Raman signal with the optimization of the grating coupling efficiency, as proposed in [17], in order to ensure an optimum overall figure-of-merit.

In this work, the square of the magnitude of the surface electric field ($|E|^2$) is computed for the fundamental TE and TM modes of a $Ta_2O_5$ slab waveguide on an $SiO_2$ substrate, with a water superstrate at a wavelength of 785 nm, and with the mode carrying 1 W power per meter slab width. A water superstrate was assumed as most future clinical or environmental analytes are expected to be aqueous. The grating coupler was designed using the process described in detail in [17]. Vectorial 2D-FDTD simulations using Lumerical FDTD Solutions™ were conducted, where a 50% duty-cycle (i.e. with the grating ridge width being half the grating period) uniform grating with etch depth $e$ and period $p$ was constructed on top of the film of thickness $h$, and a mode expansion monitor was used to capture the power of the fundamental mode coupled into the film (Fig 1). Power monitors were positioned above and below the device to

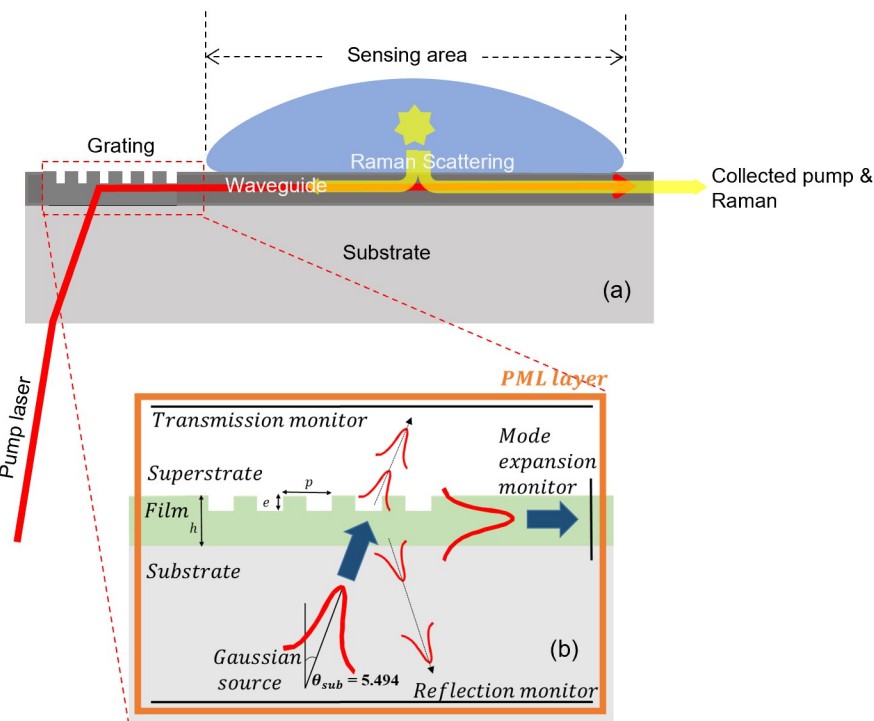

**Fig 1. (a) A schematic diagram illustrating the use of a grating coupler to launch light into a slab waveguide (b) A cross-sectional schematic diagram of the simulation layout used to model the grating coupler's performance.** *p* denotes the period (or pitch) of the grating, *e* the etch depth, and *h* the initial film thickness.

capture the amount of light transmitted straight through the grating and the light reflected from the grating back into the substrate. The optimum *p* and *e* values can be deduced either through using a parameter optimization algorithm (such as a particle swarm optimization (PSO)) or through running a nested sweep of *p* and *e* values. The method used in this work starts with running a PSO with coarse mesh parameters to quickly find the approximate optimum parameter space region. The reduced parameter space is then utilized to select the boundary values of a small nested loop with fine mesh parameters so that the exact optimum values of *p* and *e* can be located. Alternatively, an analytical approach can be adopted at the cost of reduced accuracy, as described in [20].

Fig 2 shows the results of the optimization process for the TE (Fig 2(a) & 2(b)) and TM modes (Fig 2(c) & 2(d)) for the $SiO_2$ / $Ta_2O_5$ /water waveguide at 785 nm. While Fig 2(a) & 2(c) demonstrate that a higher peak $|E|^2$ value is achieved for the TM mode than the TE mode, the maximum grating coupling efficiency into the TM mode is much lower than that into TE mode of the waveguide. This results in a higher FOM for the TE mode than the TM mode (Fig 2(b) & 2(d)), with the optimum film thickness for the TE mode being 100 nm. We optimized to a proposed figure-of-merit (FOM) which combines optimization of the square of the magnitude of the surface electric field ($|E|^2$) and the optimization of the grating coupling as follows:

$$FOM = \left( |E|^2 \right)^2 CE$$

where CE is the grating coupling efficiency.

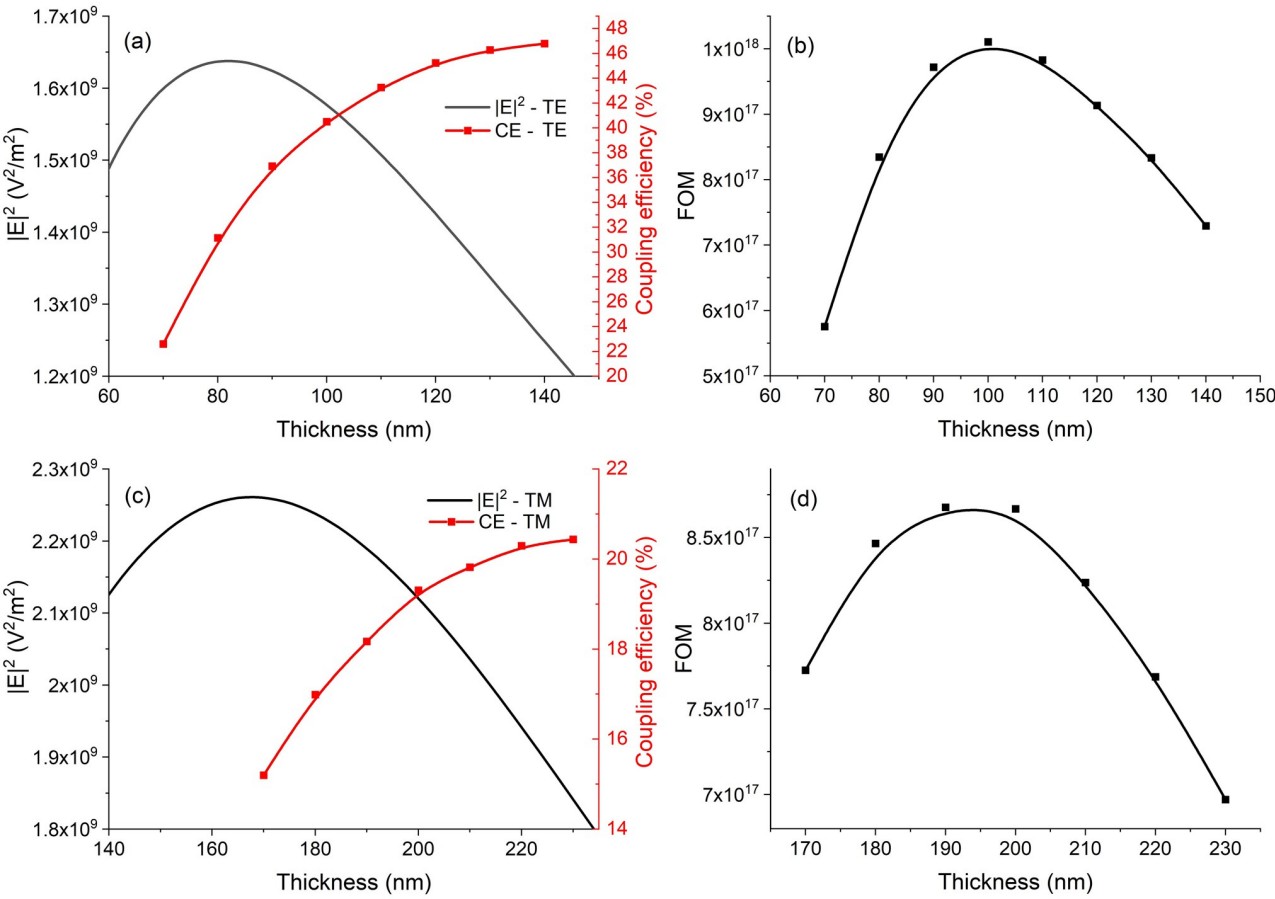

**Fig 2. $|E|^2$, grating coupling efficiency and FOM as a function of the film thickness for the fundamental TE ((a) and (b)) and TM mode ((c) and (d)).**

The optimum $p$ and $e$ values in the case of the TE mode, for the optimum 100 nm waveguide thickness deduced above, were 572.2 nm and 66.6 nm, respectively, and lead to a predicted coupling efficiency of approximately 40% at a wavelength of 785 nm.

Ta$_2$O$_5$ waveguides were deposited on 1 mm thick, 30 mm x 40 mm SiO$_2$ substrates. Following cleaning with acetone and isopropyl alcohol, substrates were immersed in a mild potassium carbonate cleaning solution (0.5 M K$_2$CO$_3$ in 3:1 methanol:water) for 30 minutes, thoroughly rinsed with de-ionized water, dried with compressed nitrogen and then baked at 120˚C in order to thoroughly dry them. Deposition of the Ta$_2$O$_5$ film was performed using an AJA International Orion Sputtering system. The radio frequency (RF) sputtering of a pure Ta$_2$O$_5$ was conducted at room temperature with a power of 250 W, and an argon to oxygen flow ratio of 20:6 sccm, with the chamber maintained at a pressure of 7 mTorr. The sputtering parameters (which include the magnetron power, the chamber pressure and temperature and argon to oxygen flow ratio) were carefully tuned to result in a minimal transmission loss and a high refractive index, and annealing the sputtered Ta$_2$O$_5$ waveguide at 550˚C for 2 hours in oxygen further reduces the loss.

Following the deposition of the Ta$_2$O$_5$ film, the designed grating coupler was written on the deposited film through e-beam lithography using an e-beam photoresist and subsequently etching the Ta$_2$O$_5$ using ion-beam milling (Oxford Instruments Ionfab 300 Plus). The e-beam

photoresist was then stripped off through plasma-ashing. Numerous demonstrations have shown the possibility of fabricating grating couplers using embossing and imprint lithography [21–23], which can further reduce the cost of making WERS chips for applications where disposability is required.

The output coupling facets were prepared by dicing, a type of mechanical sawing which is normally used for the singulation of individual die from a wafer. Through careful selection of dicing saw blade composition and cutting parameters, ductile removal of material may be achieved. Ductile dicing removes material by plastic deformation resulting in nanometer scale roughness with no cracks or chips, without the need for further polishing steps [24] for the efficient outcoupling of the Raman spectra by end-fire coupling into a large-core multimode fiber.

The fabricated waveguide with the grating coupler was characterized in terms of coupling efficiency at 785 nm, using a 1 mm X 50 μm elliptical beam size and an 8° angle of incidence, and was estimated to be 35% by measuring the power in the incident beam and at the end of the waveguide and accounting for the measured propagation loss of the waveguide. To experimentally determine tolerance to translational alignment, the waveguide was translated along the direction of propagation and the points at which the coupled power dropped to -3 dB of its peak value were found to be separated by 65 μm (Fig 3), which is a sufficiently relaxed tolerance for an alignment-free mechanical design. The reduced coupling efficiency compared to the theoretically expected figure may be due to (i) the non-ideal 50% duty cycle resulting from the e-beam process (Fig 4(b)), and (ii) the deviation from the perfectly rectangular groove profile as a result of the ion-beam milling process, which gives rise to non-vertical sidewall angles. Both of these issues can be accounted for during the modeling process of the grating coupler, and as such, their impact on the coupling efficiency can be mitigated in the future. Nevertheless, the coupling efficiency achieved is superior to previous WERS demonstrations utilizing aspheric lenses or microscope objectives, and is only surpassed by N. F. Tyndall et al.'s demonstration [14], which used monomode optical fibers end-fire coupled to sensing waveguides with inverse-taper edge couplers, and achieved a maximum coupling efficiency of 48%. While the integration of edge couplers with optical fibers is attractive for long-duration applications, it requires more complex fabrication and precise active alignment and attachment, making it unattractive for alignment-free interchange of disposable chips for clinical applications.

## 3. Selection of reference analyte

Sensors require standards for calibration and for systematic optimization of device designs. Ideally, a standard should be readily available, stable, safe to handle and have well-established, quantifiable characteristics. A standard for Raman scattering based sensors should have a well-quantified spectrum in terms of the cross-sections and frequencies of the Raman-active vibrations. Toluene is a commonly used standard as it is a small, simple, well-characterized molecule and is highly polarizable due to its aromatic ring. However, the use of toluene is problematic because it is both volatile and toxic, rendering it unsuitable for use outside a specialized laboratory. Benzyl alcohol presents as a more suitable standard analyte as it is non-volatile, has low toxicity and is readily available, with its low volatility enabling more stable long-duration on-chip measurements.

Raman spectra were taken from each analyte in Suprasil cuvettes under a Leica DMLM microscope feeding a Renishaw 2000 Raman spectrometer, using a 10 mW, 785 nm laser with an acquisition time of 10 s. Five analyte spectra were averaged and background corrected using five averaged spectra from an empty cuvette.

**Fig 3. The grating coupler's normalized coupling efficiency as a function of translational misalignment.** The squares represent the experimentally measured data, while the solid line is a polynomial fit.

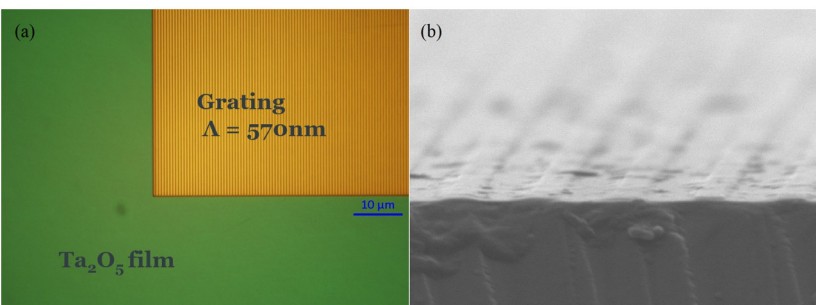

**Fig 4. (a) Scanning electron microscopy (SEM) image of the top of the Ta$_2$O$_5$ waveguide with the grating coupler, (b) SEM cross-section of the etched grating.**

The measured Raman spectra of toluene and BnOH, processed as described above, are shown in Fig 5, and exhibit similar characteristics due to the similarity of their chemical structures. The large characteristic band at 1002 cm⁻¹ in BnOH originates from the same vibrational mode as the 1003 cm⁻¹ band in toluene, and is slightly shifted due to the substitution of a different group on the benzene ring. In both cases the mode is assigned to the $\nu(C = C)$ 12 vibration of the aromatic ring using Wilson's notation [25, 26]. Absolute Raman scattering cross-sections have been measured for only a few simple molecules, including toluene, but not BnOH. However, from the spectra it is clear that the cross section for BnOH is similar to that for toluene.

The Raman spectrum of deuterated benzyl-d7 alcohol is also shown in Fig 5 as this provides a second potential standard in which the frequencies of the vibrational bands are systematically shifted. In this case, due to experimental constraints, a cuvette was not used but the apparatus was otherwise identical, so that the vibrational frequencies and relative magnitudes of the peaks are correct. In this case, the spectrum been normalized to the 1002 cm⁻¹ peak height of benzyl alcohol for ease of comparison. Deuteration is observed to shift the 1002 cm⁻¹ band

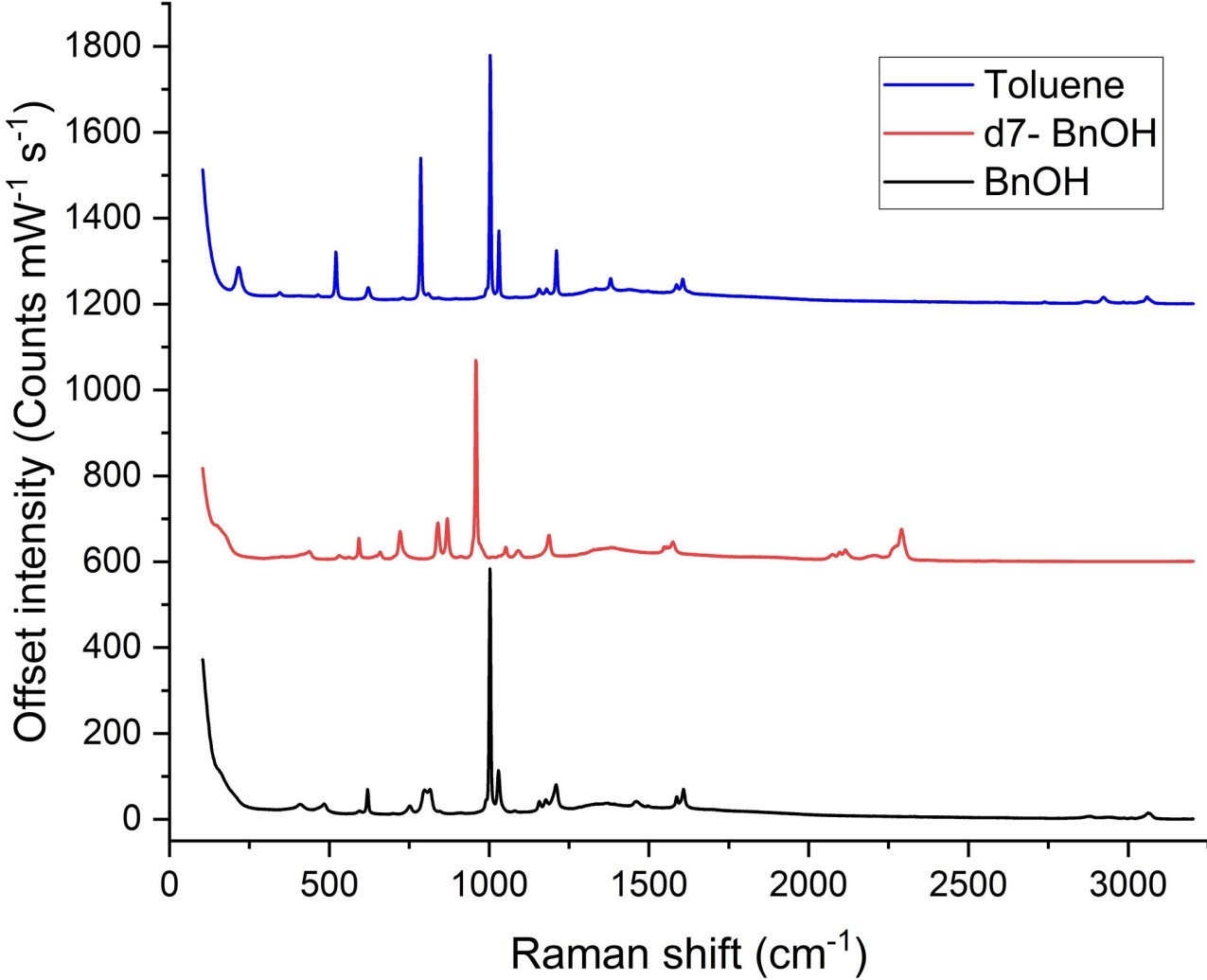

**Fig 5. Raman spectra of toluene, benzyl alcohol and d7-benzyl alcohol.** Spectra shifted up for clarity.

down to 959 cm$^{-1}$ due to the greater mass of D compared with H, a shift of 43 wavenumbers. A detailed discussion of the shifts in bond frequencies is given in [27–29].

## 4. WERS experimental apparatus and results

The experimental apparatus used for the WERS measurements is shown in Fig 6. An 8 dBm 785 nm polarization maintaining fiber-coupled volume Bragg grating (VBG) laser diode (Integrated Optics—model 785 NM SLM) is used as a pump source. The move from 633 nm pump wavelength used in our earlier demonstrations to 785 nm is due to the low water absorption and low fluorescence of biological media at this wavelength. The laser is collimated by an aspheric lens-based fiber collimator and is subsequently passed through a 785 nm Semrock MaxLine laser clean-up filter (pre-filter) to eliminate the long-wavelength laser background emission. The filtered laser beam is then directed towards a dielectric mirror that is mounted on a high-precision rotation mount to control the angle of reflection from the mirror. The reflected beam is directed towards the bottom of the $Ta_2O_5$ waveguide, which is mounted on a sample holder above the mirror (see inset of Fig 6), and travels through the substrate and film before incidence upon the grating coupler written on the top of the $Ta_2O_5$ film. The grating coupler, written in the middle of the 3 cm long chip to ensure optimum sensing length based on the propagation loss of the sample, couples light into the TE mode of the waveguide, which is then collected by a multi-mode (MM) fiber that is butt-coupled onto the end-facet of the chip. The collected pump and Raman signal are then collimated, bandpass filtered (Semrock 785 nm EdgeBasic™) to reject the pump and coupled into a second MM fiber. The filtered Raman signal is then finally passed to a compact spectrometer (Wasatch model WP-785-ER) for signal analysis.

The WERS apparatus was used to measure the Raman spectra for benzyl alcohol (BnOH) and its deuterated form (d7-BnOH). A background (reference) spectrum was first recorded to allow baseline subtraction when measuring the analyte of interest. Subsequently, a droplet of BnOH was cast on top of the waveguide covering an 18 mm length of the waveguide. The reference spectrum showed a strong Raman background in the 0–800 cm$^{-1}$ range, with a significant Raman band at 660 cm$^{-1}$ corresponding to the Ta–O stretching vibrations of $TaO_6$

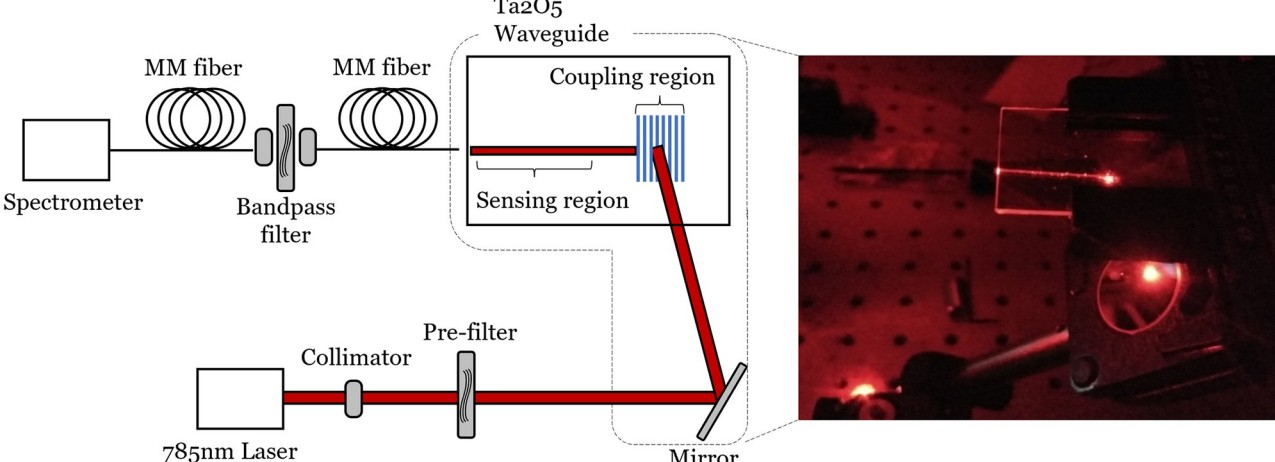

**Fig 6. The experimental apparatus used for the WERS demonstration.** Inset: illustrative photo of a 633 nm pump beam being reflected by the mirror towards the bottom of the chip and the subsequent coupling of light into the film (as indicated by the strong streak of light) by the grating coupler.

octahedra [30], while a much lower background at higher Raman shifts is observed. The raw Raman spectra recorded by the spectrometer were first normalized with respect to the maximum peak, before subtracting the background spectrum. A non-local means (NLM) denoising algorithm, which is known as an "edge preserving" (lower detail loss) denoising method in imaging, was then applied. The main feature of such an algorithm, in contrast with conventional local mean filters, is that the NLM takes the similarity-weighted mean of all pixels to filter the target pixel (further details of the NLM algorithm are reported in [31]). Subsequently, a deconvolution process was carried out, resulting in the final processed spectra of Fig 7(a). Upon casting the BnOH droplet, associated Raman bands emerge (Fig 7(a)), with the most prominent band appearing at 1002 cm$^{-1}$, followed by weaker bands at 1206 cm$^{-1}$, 1460 cm$^{-1}$,1608 cm$^{-1}$ and 3064 cm$^{-1}$. After measuring the Raman spectrum of BnOH, the BnOH droplet was removed from the top of the waveguide, and a droplet of d7-BnOH was then cast in its place. The application of the d7-BnOH droplet resulted in multiple new Raman bands appearing in the spectrum, which are shifted with respect to the aforementioned BnOH Raman bands. The first peak at 959 cm$^{-1}$ was again the most prominent, with weaker bands at

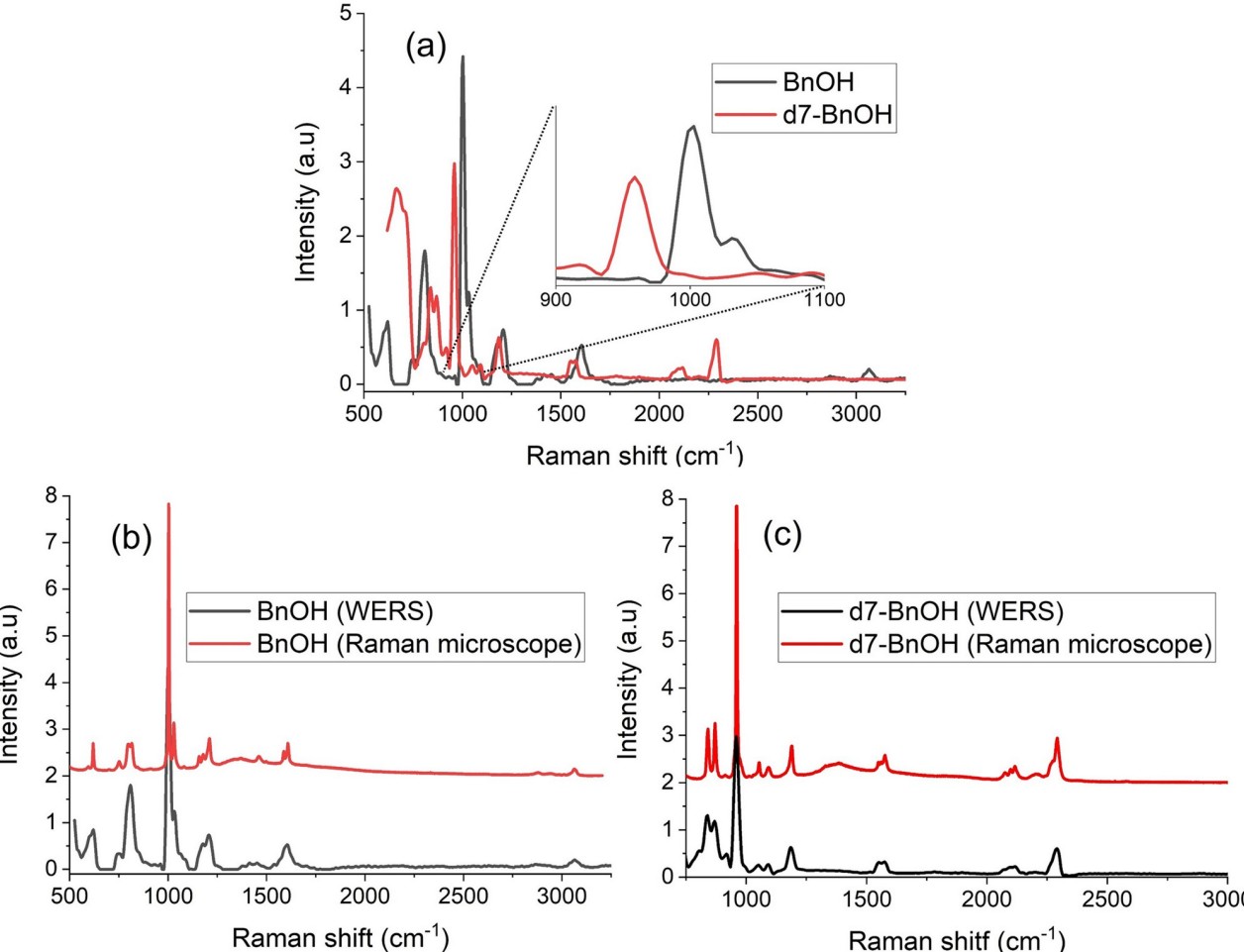

**Fig 7.** (a) The processed spectra of BnOH and d7- BnOH after background subtraction, denoising and deconvolution. Inset: Zoomed-in spectra between 900 cm$^{-1}$ & 1100 cm$^{-1}$. (b) Comparison of the WERS spectra with those measured using a commercial Raman microscope for (b) BnOH and (c) d7- BnOH. The Raman microscope spectra were scaled and shifted up for clarity.

1189 cm$^{-1}$, 1576 cm$^{-1}$, 2115 cm$^{-1}$ and 2291 cm$^{-1}$ appearing thereafter. Fig 7(b) and 7(c) compare the Raman spectra measured by the WERS system with those measured by a commercial Raman microscope for of BnOH and d7-BnOH, respectively. The figures show a good agreement between the two systems in terms of the position and relative intensity of the Raman bands and demonstrates the capability of the WERS system to detect the Raman shifts and quantify the Raman emission intensities of both BnOH and d7-BnOH.

The spectra presented in Fig 7 highlight the value of using BnOH and d7-BnOH as analytes for characterizing and benchmarking Raman systems. Unlike toluene, which is commonly used to benchmark Raman system, BnOH and its deuterated form are colorless non-volatile liquids with a mild odor and low toxicity, moreover they possess a comparable Raman cross-section to toluene, for instance, making their use more convenient and safe.

The efficiency of the WERS system was characterized by estimating the power in the 1002 cm$^{-1}$ BnOH Raman band. This was achieved by establishing the relationship between spectrometer counts and power through launching a known amount of power from a broadband white light source, filtered by a bandpass filter, into the spectrometer, which resulted in a measured spectrometer sensitivity of 140 counts/ms/pW. Following this procedure, the power in the 1002 cm$^{-1}$ BnOH Raman band was estimated to be $1 \times 10^{-14}$ W, which is a six times stronger signal than was obtainable from our previous 633 nm Ta$_2$O$_5$-based WERS system [32], utilizing a similar slab waveguide design, a prism coupling system and toluene as a target analyte, when considering the significant decrease in pump power used (44 mW of pump power was used in [32] vs. 6.3 mW here). This is despite the reduced Raman cross-section when exciting at a wavelength of 785 nm rather than 633 nm (a factor of ~2 reduction). This enhancement is a result of many factors, which include the improved coupling efficiency, the superior Raman excitation due to the optimized waveguide design, the enhanced waveguide facet quality, and the reduced fluorescence background due to operating at the longer wavelength of 785 nm.

## 5. Conclusion

Waveguide-enhanced Raman spectroscopy (WERS) is a promising technique for biochemical analysis, being potentially simpler, cheaper and more repeatable than surface-enhanced Raman spectroscopy (SERS). However, while the collection of the Raman signal can be readily achieved using a large core-diameter optical fiber, a challenge for WERS is launching light into the very thin waveguides required with high efficiency and reasonable alignment tolerance, particularly for routine use with disposable chips. In this paper we present grating-coupled WERS with through-chip excitation and alignment tolerance appropriate for simple alignment-free mechanical location, providing a direct route towards miniaturizing and packaging it as a portable WERS instrument for point-of-care and field-testing applications where disposable chips can be readily interchanged. The use of slab waveguides allows the achievement of lower losses and greater alignment and fabrication tolerances while also reducing the costs in comparison with channel waveguides, with only a small reduction in Raman conversion efficiency. Raman spectra of benzyl alcohol and deuterated benzyl alcohol, convenient for calibration and benchmarking Raman systems due to their low toxicity and volatility were measured on low-background Ta$_2$O$_5$ waveguides, clearly showing the changes in vibration frequencies on deuteration. Further improvements could be made by using apodized gratings (where the grating period and/or etch depth are varied along the grating length) which should result in improved coupling efficiency. Moreover, while e-beam lithography was used to fabricate the gratings in this work, it is important to note that costs could be reduced significantly by using embossing or imprinting for mass-production. Application of this sensing platform is proposed for a wide range of clinical measurements where disposability is required. Further, in

addition to the advantages of the grating-coupling approach, the use of $Ta_2O_5$ is attractive due to its excellent biocompatibility for sensor application [33] and as it can be readily covalently modified to attach biorecognition elements [34]. Grating coupling has not been used for WERS devices reported previously, mostly because end-fire and prism coupling are more flexible for research into the fundamentals of waveguide enhancement. We report the use of grating input coupling here in a move towards the practical deployment of alignment-tolerant disposable chips in the field, where the alignment and adjustment of lenses or prisms or the use of costly pigtailed devices is cumbersome not appropriate.

## Author Contributions

**Conceptualization:** Mohamed A. Ettabib, Almudena Marti.

**Formal analysis:** Bethany M. Bowden.

**Investigation:** Mohamed A. Ettabib.

**Methodology:** Mohamed A. Ettabib, Bethany M. Bowden, Glenn M. Churchill, James C. Gates.

**Software:** Zhen Liu.

**Supervision:** Michalis N. Zervas, Philip N. Bartlett, James S. Wilkinson.

**Visualization:** Mohamed A. Ettabib, Bethany M. Bowden, Zhen Liu.

**Writing – original draft:** Mohamed A. Ettabib.

**Writing – review & editing:** Mohamed A. Ettabib, Michalis N. Zervas, Philip N. Bartlett, James S. Wilkinson.

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
