## [Decision Letter · Decision Letter 0]

26 Jan 2023

PONE-D-22-34445Grating-Incoupled Waveguide-Enhanced Raman SensorPLOS ONE

Dear Dr. Ettabib,

Thank you for submitting your manuscript to PLOS ONE. After careful consideration, we feel that it has merit but does not fully meet PLOS ONE’s publication criteria as it currently stands. Therefore, we invite you to submit a revised version of the manuscript that addresses the points raised during the review process.

We look forward to receiving your revised manuscript.

Kind regards,

Xuejian Wu, Ph.D.

Academic Editor

PLOS ONE

Journal Requirements:

"This work was funded by a grant from the UK Engineering and Physical Sciences Research Council (EPSRC), Grant Number EP/R011230/1. 

B.M.B. thanks the Defence Science and Technology Laboratory (contract no. DSTLX-1000128554) for supporting an EPSRC industrial CASE award"

"This work was funded by a grant from the UK Engineering and Physical Sciences Research Council (EPSRC), Grant Number EP/R011230/1. B.M.B. thanks the Defence Science and Technology Laboratory (contract no. DSTLX-1000128554) for supporting an EPSRC industrial CASE award. All data supporting this study are openly available at " ext-link-type="uri" xlink:type="simple">https://doi.org/10.5258/SOTON/D1855."

"This work was funded by a grant from the UK Engineering and Physical Sciences Research Council (EPSRC), Grant Number EP/R011230/1. 

B.M.B. thanks the Defence Science and Technology Laboratory (contract no. DSTLX-1000128554) for supporting an EPSRC industrial CASE award"

6. PLOS requires an ORCID iD for the corresponding author in Editorial Manager on papers submitted after December 6th, 2016. Please ensure that you have an ORCID iD and that it is validated in Editorial Manager. To do this, go to ‘Update my Information’ (in the upper left-hand corner of the main menu), and click on the Fetch/Validate link next to the ORCID field. This will take you to the ORCID site and allow you to create a new iD or authenticate a pre-existing iD in Editorial Manager. Please see the following video for instructions on linking an ORCID iD to your Editorial Manager account: https://www.youtube.com/watch?v=_xcclfuvtxQ.

Reviewers' comments:

Reviewer's Responses to Questions

**Comments to the Author**

1. Is the manuscript technically sound, and do the data support the conclusions?

Reviewer #1: Yes

Reviewer #2: Yes

Reviewer #3: Yes

2. Has the statistical analysis been performed appropriately and rigorously? 

Reviewer #1: Yes

Reviewer #2: Yes

Reviewer #3: N/A

3. Have the authors made all data underlying the findings in their manuscript fully available?

Reviewer #1: Yes

Reviewer #2: Yes

Reviewer #3: Yes

4. Is the manuscript presented in an intelligible fashion and written in standard English?

Reviewer #1: Yes

Reviewer #2: Yes

Reviewer #3: Yes

5. Review Comments to the Author

Reviewer #1: In this paper, the authors investigate using a grating coupler for the input optical coupling for a waveguide-enhanced Raman spectroscopy (WERS) system. Substantial experiments are given, but the manuscript needs additional revisions to have a clear convey and become technically sound. My detailed comments are as follows.

1) Please consider adding a table to summarize the comparisons between SERS and WERS.

2) Please also consider adding a table to summarize the comparisons of different coupling approaches used for WERS.

3) In the introduction, the author attributes the high coupling loss of WERS to "the nature of the ultrathin film planar waveguide." Please explains this "nature" explicitly.

4) The paragraph starts with "While grating couplers are extensively ..." is hard to follow and seems irreverent. At the same time, there lacks a discussion about previous works that have demonstrated grating couplers for WERS. If there are any previously demonstrated works, please include them. If not, please discuss potential reasons why the grating couplers were not implemented before.

5) Some abbreviations are not given with full spell-outs at where they first appeared, such as "FOM" and "RF sputtering"

6) Is there any reason the FOM is proportional to the square of the magnitude of the surface electric field?

7) Please add a plot illustrating the drop in coupling efficiency and the translational misalignment.

8) Exaggerative and ambiguous words, such as numerous and impressive, should be avoided.

Reviewer #2: The authors have reported a waveguide-enhanced Raman spectroscopy (WERS) platform with alightment-tolerant under-chip grating input coupling and the design is properly tested with experiment. I recommend this paper to be published.

Reviewer #3: Waveguide-Enhanced Raman Spectroscopy (WERS) is a rapidly developing method of chemical and biological sensing. In this manuscript, Ettabib et al describe an interesting experimental followup to their 2020 Optics Express paper on the design of input grating couplers for WERS. The work is overall rigorous and well described, and will be a useful contribution to the field. I have just a few minor comments that should be addressed prior to publication.

1) Authors state that gratings aren’t needed on the output because the output is “alignment tolerant”. This is true, but it would be useful for readers who aren’t experts in photonics to have some indication of how much the tolerance is for a typical waveguide - large core multimode fiber.

2) A complete top view schematic of the chip should be provided.

3) The authors make the statement that “Numerous demonstrations have shown the possibility of fabricating grating couplers using embossing and imprint lithography”. If this has been done for gratings of similar dimensions as those described here (p = 572.2 nm, e = 66.6 nm), it would be useful to include a couple of citations.

6. PLOS authors have the option to publish the peer review history of their article (what does this mean?). If published, this will include your full peer review and any attached files.

Reviewer #1: No

Reviewer #2: No

Reviewer #3: No

---

## [Author Response · Author response to Decision Letter 0]

15 Mar 2023

Dear Professor Wu,

We would like to express our thanks to you and the reviewers for the time spent considering our manuscript. We are very pleased that all three reviewers have expressed positive opinions, shown confidence in our work and recommended our paper for publication. 

We have addressed the reviewers’ feedback and have significantly enhanced the revised manuscript to reflect the reviewers’ suggestions and comments. A detailed list of the changes made in order to address the comments and questions raised by the reviewers is outlined below.

Yours Sincerely,

Mohamed A. Ettabib 

Reviewer #1: In this paper, the authors investigate using a grating coupler for the input optical coupling for a waveguide-enhanced Raman spectroscopy (WERS) system. Substantial experiments are given, but the manuscript needs additional revisions to have a clear convey and become technically sound. My detailed comments are as follows.

1) Please consider adding a table to summarize the comparisons between SERS and WERS.

We thank the Reviewer for this suggestion. When attempting to formulate a table to compare SERS and WERS we found that the parameters for comparison are numerous, and the realisations diverse, making the table design awkward and of little additional benefit to what has already been stated in the text:

“WERS operates on the same principles as surface-enhanced Raman spectroscopy (SERS) with electromagnetic enhancement of Raman scattering at the sensing surface, and is thus expected to be useful in the same range of analytical applications as SERS [1], where it would be would be applicable to a wide range of biosensor assays, such as lateral flow immunoassays, DNA discrimination and detection, affinity assays, and biomarker detection, in the same way as SERS [5,6]. Similar approaches would also be adopted for the suppression of interferences such as surface modification to suppress non-specific adsorption [7]. However, WERS sensors offer numerous advantages over the more mature SERS technology, as they are low cost, offer a high degree of miniaturization, and therefore enhanced practicality and usability, and unlike SERS, do not utilize expensive and fragile noble metal nanostructures. Furthermore, unlike SERS, the optical excitation and collection light-paths do not pass through the liquid sample volume, reducing interference from absorption and scattering, also removing the need for an optical window for light to enter and leave the sample and allowing direct placement of fluidics on the waveguide.”

We feel that in this case this text along with a citation to our WERS review paper, which discusses extensively the differences between WERS and SERS, makes a more representative comparison than a table and so would prefer to omit it.

2) Please also consider adding a table to summarize the comparisons of different coupling approaches used for WERS.

We agree with the Reviewer that including a table that compares the of different coupling approaches used for WERS would be beneficial. We have therefore tabulated the main differences between the four main coupling methodologies used for WERS, namely, prism coupling, grating couplers, end-fire coupling (using Aspheric lenses / Microscope objective lenses) and fiber pigtailing:

 Coupling efficiency Bandwidth Alignment Tolerance Cost Complexity

Prism coupling 10-30% Moderate; tens of nanometres Good; tens of microns Moderate Moderate. Requires prism attachment

Grating couplers 30-50% Moderate; tens of nanometres Good; tens of microns Moderate Moderate. Requires a nanofabrication facility

Aspheric lenses / Microscope objective lenses 10-30% High; hundreds of nanometres. Poor; a few microns Moderate Easy to set-up

Fiber pigtailing with inverse tapers 50% High; hundreds of nanometres Good – once fibres are fixed to the chip High Complex. Requires a nanofabrication facility and precise active alignment and attachment

The table can be found in Page 3, Line 7 of the revised manuscript. 

3) In the introduction, the author attributes the high coupling loss of WERS to "the nature of the ultrathin film planar waveguide." Please explains this "nature" explicitly.

We thank the Reviewer for allowing us to expand on this point. Waveguides fully optimized for surface sensing are ultrathin (�0.1 µm) and support only one mode in each polarization, so that the field distribution and modal velocity is fixed, leading to highly quantifiable operation and great stability. In addition, the ultrathin thickness serves to generate a strong evanescent field through which laser light interacts with analytes residing on the surface of the waveguide. We have added the following text to the revised manuscript (Page 2, Line 14): 

“…, where end-coupling to ultrathin high index contrast waveguides is made difficult by their small mode size (�1 µm FWHM) which requires a similar incident spot/mode for good coupling efficiency and submicron alignment tolerances, or alternatively integrated mode transformers which still demonstrate micron-scale alignment tolerances.”

4) The paragraph starts with "While grating couplers are extensively ..." is hard to follow and seems irreverent. At the same time, there lacks a discussion about previous works that have demonstrated grating couplers for WERS. If there are any previously demonstrated works, please include them. If not, please discuss potential reasons why the grating couplers were not implemented before.

While we agree with the Reviewer that a discussion about previous works that have demonstrated grating couplers for WERS would be interesting, unfortunately this is not possible as our work is the first to demonstrate the use of gratings couplers for launching light into a WERS sensor. 

However, we share the Reviewer’s opinion that a brief discussion of the possible reasons of the lack of adoption of grating couplers as a coupling mechanism for WERS should be had. We think that there are two main reasons: 

(1) While the WERS principle was first demonstrated as early as 1972, the recent re-ignited research interest in WERS has been less than a decade long, fuelled by the advent of low-cost laser diodes, compact spectrometers and recent progress in material engineering, nanofabrication techniques and software modelling tools that have made realising portable and cheap WERS Raman systems with high sensitivity a realistic possibility. This makes the WERS field of limited relative maturity compared, for instance, to SERS, and as such, there are numerous waveguide designs, coupling methods, material platforms and potential applications still yet to be explored. Using prisms, microscope objective lenses and butt-coupled optical fibres are straightforward demountable techniques for exploratory studies, but in this paper we aim to move closer to a practically-deployable disposable device, which requires no complex assembly or manipulation and relaxed alignment tolerances.

(2) While some coupling methods such as prism coupling and end-fire coupling using objective lenses are techniques that are easily implementable and require components that are widely commercially available, grating couplers require a nanofabrication facility with access to e-beam lithography to pattern them, which is more costly and less readily accessible than the aforementioned methods. 

Our paper is the first to report a grating-incoupled WERS sensor, where we show that the advantages attained in terms of simplicity, flexibility and tolerance make grating couplers the best candidate for a wide range of WERS applications, particularly those requiring alignment-free interchange of disposable chips. Furthermore, while the grating coupling efficiencies are, as expected, lower than what is achievable in silicon photonics, we show that they are still higher than what has been achieved with alternative WERS coupling methods. By reporting such progress, we hope that future research can further improve coupling efficiencies as well as demonstrate alternatives routes towards fabricating the grating couplers without the need for e-beam lithography, such as through the use of embossing or imprinting, which would significantly reduce the cost. 

We have added the following text to the revised manuscript (Page 12, Line 28): 

“Grating coupling has not been used for WERS devices reported previously, mostly because end-fire and prism coupling are more flexible for research into the fundamentals of waveguide enhancement. We report the use of grating input coupling here in a move towards the practical deployment of alignment-tolerant disposable chips in the field, where the alignment and adjustment of lenses or prisms or the use of costly pigtailed devices is cumbersome and not appropriate.”

5) Some abbreviations are not given with full spell-outs at where they first appeared, such as "FOM" and "RF sputtering"

We thank the Reviewer for bringing this to our attention, which we have now addressed in the revised manuscript. 

6) Is there any reason the FOM is proportional to the square of the magnitude of the surface electric field?

The proposed FOM given in our manuscript is FOM = (|E|2) 2 CE where CE is the grating coupling efficiency. |E|2 is used instead of the intensity, I, because the evanescent electric field may contain a component in the direction of modal propagation (as described in [2]). The excitation of a molecule is proportional to |E|2 and the emission into the waveguide mode is also proportional to |E|2 . The choice of squaring |E|2 is made to include both the waveguide excitation and collection of the Raman signal, under the conventional assumption that the Stokes shift is small. We have added the following text (Page 4, 

Line 5) to explain this: 

“…. [2] as the evanescent electric field may contain a component in the direction of modal propagation (e.g. in the case of TM polarization)”

7) Please add a plot illustrating the drop in coupling efficiency and the translational misalignment.

We thank the Reviewer for their suggestion. We agree that a plot conveying the relationship between the CE and translational misalignment would be useful to the reader and have thus added it in the revised manuscript. 

Fig. 3. The grating coupler’s normalized coupling efficiency as a function of translational misalignment. The squares represent the experimentally measured data, while the solid line is a polynomial fit. 

The plot can be found in Page 7, Line 1 of the revised manuscript. 

8) Exaggerative and ambiguous words, such as numerous and impressive, should be avoided.

We agree with the Reviewer that the use of these words could be misunderstood and convey the wrong impression. We have toned down the language in the following sentences following the Reviewer’s suggestion:

Original: “WERS sensors offer numerous advantages over the more mature SERS technology”

Revised: “WERS sensors offer a number of advantages over the more mature SERS technology”

Original: “These advantages have recently led to a series of impressive demonstrations”

Revised: “These advantages have recently led to a series of useful demonstrations”

Reviewer #2: The authors have reported a waveguide-enhanced Raman spectroscopy (WERS) platform with alightment-tolerant under-chip grating input coupling and the design is properly tested with experiment. I recommend this paper to be published.

We thank the Reviewer for the faith they have shown in our paper, and we hope that they find the revised manuscript an even more rounded piece of work. 

Reviewer #3: Waveguide-Enhanced Raman Spectroscopy (WERS) is a rapidly developing method of chemical and biological sensing. In this manuscript, Ettabib et al describe an interesting experimental followup to their 2020 Optics Express paper on the design of input grating couplers for WERS. The work is overall rigorous and well described, and will be a useful contribution to the field. I have just a few minor comments that should be addressed prior to publication.

1) Authors state that gratings aren’t needed on the output because the output is “alignment tolerant”. This is true, but it would be useful for readers who aren’t experts in photonics to have some indication of how much the tolerance is for a typical waveguide - large core multimode fiber.

We thank the reviewer for this suggestion to make the paper more quantitative. We have added the following text to the revised manuscript (Page 2, Line 34): 

“For example, the translational alignment tolerance (defined as 3dB reduction in coupling efficiency) for coupling from an ultrathin slab waveguide into a 1mm core diameter high NA fibre is estimated to be ±0.4 mm.”

2) A complete top view schematic of the chip should be provided.

We thank the Reviewer for this suggestion. While we agree that it is often the case that a top view schematic of a chip is a useful inclusion, in our case, the simple nature of our chip, which is simply a slab waveguide with a grating, is rather unexciting and does not convey much useful information beyond what is already available or inferable from Fig. 1, Fig. 4, Fig. 6 and the text. We include here a top view schematic to illustrate our point.

3) The authors make the statement that “Numerous demonstrations have shown the possibility of fabricating grating couplers using embossing and imprint lithography”. If this has been done for gratings of similar dimensions as those described here (p = 572.2 nm, e = 66.6 nm), it would be useful to include a couple of citations.

We agree with the Reviewer that citations on fabricating grating couplers using embossing and imprint lithography would be useful to the reader. We have included the following references to address this point: 

21. P. Karasiński, “Embossable grating couplers for planar evanescent wave sensors,” Opto-Electron. Rev. 19, 10–21 (2011).

22. S. Scheerlinck, D. van Thourhout and R. Baets, “UV-based Nano Imprint Fabrication of Gold Grating Couplers on Silicon-on-Insulator,” Digest of the IEEE/LEOS Summer Topical Meetings, Portland, OR, USA, 82-83 (2007).

23. S. Scheerlinck, R. H. Pedersen, P. Dumon, W. Bogaerts, U. Plachetka, D. Van Thourhout, R. Baets, A. Kristensen, “Fabrication of nanophotonic circuit components by thermal nano imprint lithography,” Conference on Lasers and Electro-Optics and Conference on Quantum Electronics and Laser Science, San Jose, CA, USA, 1-2, (2008)

---

## [Editor Report · Decision Letter 1]

22 Mar 2023

Grating-Incoupled Waveguide-Enhanced Raman Sensor

PONE-D-22-34445R1

Dear Dr. Ettabib,

We’re pleased to inform you that your manuscript has been judged scientifically suitable for publication and will be formally accepted for publication once it meets all outstanding technical requirements.

Kind regards,

Xuejian Wu, Ph.D.

Academic Editor

PLOS ONE
---

## [Editor Report · Acceptance letter]

12 Apr 2023

PONE-D-22-34445R1 

Grating-Incoupled Waveguide-Enhanced Raman Sensor 

Dear Dr. Ettabib:

I'm pleased to inform you that your manuscript has been deemed suitable for publication in PLOS ONE. Congratulations! Your manuscript is now with our production department. 

Kind regards, 

on behalf of

Dr. Xuejian Wu 

Academic Editor

PLOS ONE